# Classifying COVID-19 hospitalizations in epidemiology cohort studies: The C4R study

Elizabeth C. Oelsner[1]*, Akshaya Krishnaswamy[1], Rafail Rustamov[2], Pallavi P. Balte[1], Tauqeer Ali[3], Norrina B. Allen[4], Howard F. Andrews[5], Pramod Anugu[6], Alexander Arynchyn[7], Lori A. Bateman[8], Jianwen Cai[8], Harry Chang[1], Lucas Chen[9], Mitchell S. V. Elkind[10,11], James S. Floyd[12,13], Kelley Pettee Gabriel[14], Sina A. Gharib[15], Jose D. Gutierrez[10], Karen Hinckley Stukovsky[16], Virginia J. Howard[14], Carmen R. Isasi[17], Lauren Jager[16], Ling Jin[9], Suzanne E. Judd[18], Alka M. Kanaya[19], Namratha R. Kandula[4,20], Maureen R. Kelly[1], Sadiya S. Khan[21], Anna Kucharska-Newton[22], Joyce S. Lee[23], Emily B. Levitan[14], Cora E. Lewis[14], Barry J. Make[24], Kimberly Malloy[3], Jennifer J. Manly[10], David Mauger[25], Yuan-I Min[6], Joanne M. Murabito[9,26], Charles G. Murphy[27], Arnita F. Norwood[6], George T. O'Connor[9,26], Victor E. Ortega[28], Ashmi A. Patel[1], Amber Pirzada[29], Elizabeth A. Regan[30], Kimberly B. Ring[8], Wayne D. Rosamond[22], David A. Schwartz[23], James M. Shikany[7], Daniela Sotres-Alvarez[8], Cheryl Tarlton[3], Janis Tse[9], Elman M. Urbina Meneses[31], Maya Vankineni[1], Sally E. Wenzel[32], Prescott G. Woodruff[33], Vanessa Xanthakis[9,34], Ji Hyun Yang[9,35], Neil A. Zakai[36,37], Ying Zhang[3], Wendy S. Post[38]

1 Division of General Medicine, Department of Medicine, Columbia University Irving Medical Center, New York, New York, United States of America, 2 Department of Medicine, Nassau University Medical Center, East Meadow, New York, United States of America, 3 Center for American Indian Health Research, Department of Biostatistics and Epidemiology, Hudson College of Public Health, University of Oklahoma Health Sciences Center, Oklahoma City, Oklahoma, United States of America, 4 Center for Epidemiology and Population Health, Department of Preventive Medicine, Northwestern University, Chicago, Illinois, United States of America, 5 Data Coordinating Center and Biostatistics Department, Mailman School of Public Health, Columbia University Irving Medical Center, New York, New York, United States of America, 6 Department of Medicine University of Mississippi Medical Center, Jackson, Mississippi, United States of America, 7 Division of General Internal Medicine and Population Science, Heersink School of Medicine, University of Alabama at Birmingham, Birmingham, Alabama, United States of America, 8 Collaborative Studies Coordinating Center, Department of Biostatistics, Gillings School of Global Public Health, University of North Carolina at Chapel Hill, Chapel Hill, North Carolina, United States of America, 9 NHLBI's Framingham Heart Study, Framingham, Massachusetts, United States of America, 10 Department of Neurology, Vagelos College of Physicians and Surgeons, Columbia University, New York, New York, United States of America, 11 Department of Epidemiology, Mailman School of Public Health, Columbia University, New York, New York, United States of America, 12 Department of Epidemiology, School of Public Health, University of Washington, Seattle, Washington, United States of America, 13 Division of General Medicine, Department of Medicine, University of Washington, Seattle, Washington, United States of America, 14 Department of Epidemiology, University of Alabama at Birmingham, Birmingham, Alabama, United States of America, 15 Division of Pulmonary, Critical Care and Sleep Medicine, University of Washington, Seattle, Washington, United States of America, 16 Department of Biostatistics, School of Public Health, University of Washington, Seattle, Washington, United States of America, 17 Department of Epidemiology and Population Health, Albert Einstein College of Medicine, Bronx, New York, United States of America, 18 Department of Biostatistics, University of Alabama at Birmingham, Birmingham, Alabama, United States of America, 19 Department of Medicine, University of California San Francisco, San Francisco, California, United States of America, 20 Division of General Internal Medicine, Department of Medicine, Northwestern University, Chicago, Illinois, United States of America, 21 Division of Cardiology, Department of Medicine, Northwestern University, Chicago, Illinois, United States of America, 22 Department of Epidemiology, Gillings School of Global Public Health, University of North Carolina at Chapel Hill, Chapel Hill, North Carolina, United States of America, 23 Division of Pulmonary Sciences and Critical Care, Department of Medicine, University of Colorado Anschutz Medical Campus, Aurora, Colorado, United States of America, 24 Division of Pulmonary, Critical Care & Sleep Medicine, Department of Medicine, National Jewish Health, Denver, Colorado, United States of America, 25 Division of Biostatistics and Bioinformatics, Department of Public Health Sciences, Pennsylvania State University, State College, Pennsylvania, United States of America, 26 Department of Medicine, Boston University Chobanian and Avedisian School of Medicine, Boston, Massachusetts, United States of America, 27 Division of Pulmonary Allergy, and Critical Care Medicine, Department of Medicine, Columbia University Medical Center, New York, New York, United States of America, 28 Division of



**Data Availability Statement:** To analyze de-identified individual participant data, researchers must receive authorization for access. This authorization is required to maintain the integrity of

the data and protect participant privacy. Those wishing to access C4R data will need to submit a Data Access Request (DAR) through the database of Phenotypes and Genotypes (dbGaP) (https://www.ncbi.nlm.nih.gov/gap/). Authorized researchers can then access C4R data from BioData Catalyst® (BDC) (https://biodatacatalyst.nlhbi.nih.gov/). BioData Catalyst (BDC) is a cloud-based research ecosystem that permits access to and a platform for the analysis of scientific data supported by NHLBI. BDC currently hosts C4R datasets from 11 of the 14 C4R cohorts with additional C4R Data becoming accessible in the coming months. Cohort data from many of the C4R parent studies is also accessible through a dbGaP data access request. BDC provides instructions for finding and viewing C4R and cohort data as well as the steps to request authorization to use de-identified individual participant data for scientific analysis within BDC. Accession numbers are XXXX.

**Funding:** The Collaborative Cohort of Cohorts for COVID-19 Research (C4R) Study is supported by National Heart, Lung, and Blood Institute (NHLBI)–Collaborating Network of Networks for Evaluating COVID-19 and Therapeutic Strategies (CONNECTS)/Researching COVID to Enhance Recovery (RECOVER) grant OT2HL156812, with co-funding from the National Institute of Neurological Disorders and Stroke (NINDS) and the National Institute on Aging (NIA). The Principal Investigator is ECO. Additional funding for participating cohorts is as follows: ARIC The Atherosclerosis Risk in Communities Study has been funded in whole or in part by the NHLBI, National Institutes of Health (NIH), US Department of Health and Human Services, under contracts 75N92022D00001, 75N92022D00002, 75N92022D00003, 75N92022D00004, and 75N92022D00005. Neurocognitive data are collected under grants U01 2U01HL096812, 2U01HL096814, 2U01HL096899, 2U01HL096902, and 2U01HL096917 from the NHLBI, the NINDS, the NIA, and the National Institute on Deafness and Other Communication Disorders. The authors thank the staff and participants of the ARIC Study for their important contributions CARDIA The Coronary Artery Risk Development in Young Adults Study (CARDIA) is conducted and supported by the National Heart, Lung, and Blood Institute (NHLBI) in collaboration with the University of Alabama at Birmingham (75N92023D00002 & 75N92023D00005), Northwestern University (75N92023D00004), University of Minnesota (75N92023D00006), and Kaiser Foundation Research Institute (75N92023D00003). This manuscript has been reviewed by CARDIA for

Pulmonary, Critical Care, Allergy, and Immunologic Diseases, Department of Medicine, Mayo Clinic Scottsdale, Scottsdale, Arizona, United States of America, **29** Institute for Minority Health Research, University of Illinois Chicago, Chicago, Illinois, United States of America, **30** Division of Rheumatology, Department of Medicine, National Jewish Health, Denver, Colorado, United States of America, **31** Division of Pulmonary and Critical Care, Graduate School of Medicine, University of Tennessee, Knoxville, Tennessee, United States of America, **32** Department of Environmental and Occupational Health, School of Public Heath, University of Pittsburgh, Pittsburgh, Pennsylvania, United States of America, **33** Division of Pulmonary, Critical Care, Allergy, and Sleep Medicine, Department of Medicine, University of California San Francisco, San Francisco, California, United States of America, **34** Department of Medicine, Boston University Chobanian and Avedisian School of Medicine, and Department of Biostatistics, School of Public Health, Boston University, Boston, Massachusetts, United States of America, **35** Division of Cardiovascular Medicine, Department of Medicine, Lahey Hospital and Medical Center, Burlington, Massachusetts, United States of America, **36** Department of Medicine, Larner College of Medicine at the University of Vermont, Burlington, Vermont, United States of America, **37** Department of Pathology & Laboratory Medicine, Larner College of Medicine, University of Vermont, Burlington, Vermont, United States of America, **38** Division of Cardiology, Department of Medicine, Johns Hopkins University, Baltimore, Maryland, United States of America

* Eco7@cumc.columbia.edu

# Abstract

## Rationale

Robust COVID-19 outcomes classification is important for ongoing epidemiology research on acute and post-acute COVID-19 conditions. Protocolized medical record review is an established method to validate endpoints for clinical trials and cardiovascular epidemiology cohorts; however, a protocol to adjudicate hospitalizations for COVID-19 among epidemiology cohorts was lacking.

## Objectives

We developed a protocol to ascertain and adjudicate hospitalized COVID-19 across a meta-cohort of 14 US prospective cohort studies. This report describes the first three years of protocol implementation (October 1, 2020—October 1, 2023) and evaluates its repeatability and performance compared to classification by administrative codes.

## Methods

The protocol was adapted from cohort approaches to clinical cardiovascular events ascertainment and adjudication. Potential COVID-19 hospitalizations and deaths were identified by self-/proxy-report and, in some cases, active surveillance. Medical records were requested from hospitals and adjudicated for COVID-19 outcomes by clinically trained personnel according to a standardized rubric. Inter-rater agreement was assessed. The sensitivity and specificity of discharge diagnosis codes was compared to adjudicated diagnoses.

## Measurements and main results

The study obtained medical records for 1,167 potential COVID-19 hospitalizations, which underwent protocolized adjudication. Adjudication confirmed COVID-19 infection was present for 1,030 (88%) events, of which COVID-19 was not the cause of hospitalization for 78 (8%). Of 952 hospitalizations determined by adjudicators to be caused by COVID-19, 319 (34%) participants were critically ill and 210 (22%) died. Pneumonia was confirmed in 822

scientific content. COPDGene The Genetic Epidemiology of COPD (COPDGene) Study was supported by awards U01 HL089897 and U01 HL089856 from the NHLBI. COPDGene is also supported by the COPD Foundation through contributions made to an industry advisory board comprised of AstraZeneca AB (Cambridge, United Kingdom), Boehringer-Ingelheim (Ingelheim am Rhein, Germany), Genentech, Inc. (South San Francisco, California), GlaxoSmithKline plc (London, United Kingdom), Novartis International AG (Basel, Switzerland), Pfizer, Inc. (New York, New York), Siemens AG (Berlin, Germany), and Sunovion Pharmaceuticals Inc. (Marlborough, Massachusetts). FHS The Framingham Heart Study has received support from the NHLBI (grant N01-HC-25195, contract HHSN268201500001I, and grant contract 75N92019D00031). HCHS/SOL The Hispanic Community Health Study/Study of Latinos (HCHS/SOL) is a collaborative study supported by contracts between the NHLBI and the University of North Carolina (contract HHSN2682013000011/N01-HC-65233), the University of Miami (contract HHSN268201300004I/N01-HC-65234), Albert Einstein College of Medicine (contract HHSN268201300002I/N01-HC-65235), the University of Illinois at Chicago (contract HHSN268201300003I/N01-HC-65236 (Northwestern University)), and San Diego State University (contract HHSN268201300005I/N01-HC-65237). The following institutes/centers/offices have contributed to the HCHS/SOL through a transfer of funds to the NHLBI: the National Institute on Minority Health and Health Disparities, the National Institute on Deafness and Other Communication Disorders, the National Institute of Dental and Craniofacial Research, the National Institute of Diabetes and Digestive and Kidney Diseases, the NINDS, and the NIH Office of Dietary Supplements. JHS The Jackson Heart Study is supported by and conducted in collaboration with Jackson State University (contract HHSN268201800013I), Tougaloo College (contract HHSN268201800014I), the Mississippi State Department of Health (contract HHSN268201800015I), the University of Mississippi Medical Center (contracts HHSN268201800010I, HHSN268201800011I, and HHSN268201800012I), the NHLBI, and the National Institute on Minority Health and Health Disparities. MASALA The Mediators of Atherosclerosis in South Asians Living in America (MASALA) Study was supported by grant R01HL093009 from the NHLBI, the National Center for Research Resources, and the National Center for Advancing Translational Sciences, NIH, through University of California, San Francisco–Clinical and

(86%) and acute kidney injury in 350 (37%); other cardiovascular and thrombotic complications were rare (2–5%). Interrater reliability among adjudicators was high (kappa = 0.85–1.00) except for myocardial infarction (kappa = 0.60). Compared to adjudication, sensitivity of discharge diagnosis codes was higher for pneumonia (84%) and pulmonary embolism (81%) than for other complications (48–70%).

## Conclusions

Protocolized adjudication confirmed four out of five COVID-19 hospitalizations in a US meta-cohort and confirmed cases of pneumonia, pulmonary embolism, and other conditions that were not indicated by discharge diagnosis codes. These results highlight the importance of validating health outcomes for use in research on COVID-19 and post-COVID-19 conditions, and some limitations of claims-based data.

## Introduction

Coronavirus disease 2019 (COVID-19) has been a leading cause of hospitalizations and deaths since 2020 [1, 2]. Large population-based studies with accurate characterization of hospitalized COVID-19 and its acute complications–as well as clinical, biomarker, and lifestyle data from before and after infection–are urgently needed to understand mechanisms of acute disease and post-acute sequelae of COVID-19 (PASC), which impacts up to 57% of hospitalized COVID-19 patients [3].

Many extant US cohort studies have collected extensive data on clinical and subclinical diseases and their risk factors, including behavior, cognition, biomarkers, and social determinants of health. Since enrollment for these studies was completed prior to the COVID-19 pandemic, they offer a unique opportunity to study risk factors for incident COVID-19 while minimizing the referral, survival, and recall biases that are common to COVID-19 case series and disease-based studies. However, in the absence of a US national healthcare system, most longstanding US epidemiology studies are not able to establish comprehensive linkages to electronic health records (EHRs) and have depended on medical records ascertainment and adjudication for cardiovascular and respiratory health outcomes. A protocol to define hospitalizations for COVID-19 among epidemiology cohorts was lacking.

The Collaborative Cohort of Cohorts for COVID-19 Research (C4R) study is a unique meta-cohort of NIH-funded prospective, observational epidemiology cohort studies that was funded to perform standardized, prospective ascertainment of COVID-19, including physician adjudication of medical records for COVID-19 hospitalizations and deaths (hereafter, "events"). The primary purpose of this report is to describe the C4R events adjudication protocol and preliminary experience applying the protocol from February 2021 to June 2023. This report includes a comparison of ICD-based versus physician adjudicated protocol-based classification for COVID-related fatal and non-fatal hospitalizations.

## Materials and methods

### Study design

C4R enrolled adult participants from 14 long-standing cardiovascular, neurological, and respiratory cohorts [4]: Atherosclerosis Risk in Communities (ARIC) [5, 6], Coronary Artery Risk Development in Young Adults (CARDIA) [7], Genetic Epidemiology of COPD (COPDGene)

Translational Science Institute grant UL1RR024131. MESA The Multi-Ethnic Study of Atherosclerosis (MESA) and the MESA SNP Health Association Resource (SHARe) are conducted and supported by the NHLBI in collaboration with the MESA investigators. Support for MESA is provided by grants and contracts 75N92020D00001, HHSN268201500003I, N01-HC-95159, 75N92020D00005, N01-HC-95160, 75N92020D00002, N01-HC-95161, 75N92020D00003, N01-HC-95162, 75N92020D00006, N01-HC-95163, 75N92020D00004, N01-HC-95164, 75N92020D00007, N01-HC-95165, N01-HC-95166, N01-HC-95167, N01-HC-95168, N01-HC-95169, R01-HL077612, R01-HL093081, R01-HL130506, R01-HL127028, R01-HL127659, R01-HL098433, R01-HL101250, and R01-HL135009 from the NHLBI; grant R01-AG058969 from the NIA; and grants UL1-TR-000040, UL1-TR-001079, and UL1-TR-001420 from the National Center for Advancing Translational Sciences. NHLBI Pooled Cohorts Study The NHLBI Pooled Cohorts Study was supported by grants R21HL153700, K23HL130627, R21HL129924, and R21HL121457 from the NIH/NHLBI. NOMAS The Northern Manhattan Study was supported by grants R01 NS29993 and R01 NS48134 from the NINDS and grant R01 AG066162 from the NIA. PrePF The Prevent Pulmonary Fibrosis cohort study was established in 2000 and has been supported by NIH awards Z01-ES101947, R01-HL095393, RC2-HL1011715, R21/33-HL120770, R01-HL097163, Z01-HL134585, UH2/3-HL123442, P01-HL092870, UG3/UH3-HL151865, and DoD W81XWH-17-1-0597. REGARDS The REeasons for Geographic and Racial Differences in Stroke (REGARDS) Study is supported by cooperative agreement U01 NS041588, co-funded by the NINDS and the NIA. Additional funding was provided by R01 HL80477 and R01 HL165452 from the NHLBI. SARP Research by the principal and co–principal investigators of the Severe Asthma Research Program was funded by the NIH/NHLBI (grants U10 HL109164, U10 HL109257, U10 HL109146, U10 HL109172, U10 HL109250, U10 HL109168, U10 HL109152, and U10 HL109086). Additional support was provided through industry partnerships with the following companies: AstraZeneca, Boehringer-Ingelheim, Genentech, GlaxoSmithKline, MedImmune, Inc. (Gaithersburg, Maryland), Novartis, Regeneron Pharmaceuticals, Inc. (Tarrytown, New York), Sanofi S.A. (Paris, France), and Teva Pharmaceuticals USA (North Wales, Pennsylvania). Spirometers used in Severe Asthma Research Program III were provided by nSpire Health, Inc.

[8], Framingham Heart Study (FHS) [9], Hispanic Community Health Study/Study of Latinos (HCHS/SOL) [10–13], Jackson Heart Study (JHS) [14–16], Mediators of Atherosclerosis in South Asians Living in America (MASALA) [17, 18], Multi-Ethnic Study of Atherosclerosis (MESA) [19], Northern Manhattan Study (NOMAS) [20], Prevent Pulmonary Fibrosis (PrePF) [21], REeasons for Geographic and Racial Differences in Stroke (REGARDS) [22], Severe Asthma Research Program (SARP) [23], Subpopulations and Intermediate Outcome Measures in COPD Study (SPIROMICS) [24], and the Strong Heart Study (SHS) [25, 26]. Details on each of the component cohorts are provided in S1 Appendix and S1 Table. Cohort participants previously consented to in-person, telephone, and/or e-mail contact and for abstraction of medical records.

C4R received funding in 2020 to perform standardized prospective data collection on COVID-19 and to harmonize pre-pandemic deep phenotyping available in the cohorts. Columbia University developed the standard questionnaires and protocols and served as the Data Coordination and Harmonization Center (DCHC) for C4R (Columbia University Institutional Review Board, AAAT3035). Of note, in some cohorts, COVID-19 questionnaires were initiated before the development of a standard C4R questionnaire; these questionnaires were later harmonized with the C4R questionnaire and classified as C4R questionnaires.

As previously reported [4], following a cohort ancillary studies model, researchers in each cohort study were directly responsible for accomplishing data collection in accordance with the standard protocols and under the supervision of their own observational studies monitoring board, steering committee, institutional review board (IRB), and any other applicable regulatory authorities. Columbia University served as the Data Coordination and Harmonization Center (DCHC) for C4R (Columbia University Institutional Review Board, AAAT3035). A full list of cohort IRBs supervising implementation of the C4R protocols is provided in S2 Table.

Following cohort-specific IRB approval and consent processes (including verbal, remote, and traditional written informed consent), adult participants in the pre-existing cohort studies were enrolled into the C4R ancillary study on a rolling basis from April 9, 2020, to February 28, 2023. COVID-19 data collection was accomplished by two waves of questionnaires (Wave 1: April 2020–May 2022; Wave 2: February 2021–February 2023), SARS-CoV-2 serosurvey (February 2021–February 2023), and ascertainment and adjudication of COVID-19 events, which are the subject of this report.

For the purposes of the work presented in this manuscript, data were accessed at Columbia University from July 1, 2021, to October 1, 2023.

## Protocol development and coordination

The C4R events protocol is included in S2 Appendix. The protocol was developed by investigators and cohort personnel with experience in clinical events ascertainment [5, 7, 9, 11, 12, 17, 19, 20, 26–33] and approved by the C4R Cohort Coordinating Committee (CCC) in February 2021. Relevant administrative and review forms were coded into the C4R Events REDCap [34, 35] toolkit, which was made available to the cohorts via a central (Columbia) instance or for local adaptation by cohort data coordinating centers (DCCs). For the central instance, cohort-specific Data Access Groups (DAGs) ensured that each cohort was only able to enter or access its own participant data. Cohorts provided bimonthly tracking reports regarding cohort-specific elements of protocol completion to the DCHC, which integrated these reports and generated status reports to the Events Subcommittee, the CCC, and funding agencies. De-identified COVID-19 events data were made available for analysis on the C4R Analysis Commons and shared with cohort-specific DCCs for cohort use and transfer to other data sharing platforms.

(Longmont, Colorado). SPIROMICS The Subpopulations and Intermediate Outcome Measures in COPD Study (SPIROMICS) has been funded by contracts with the NIH/NHLBI (contracts HHSN268200900013C, HHSN268200900014C, HHSN268200900015C, HHSN268200900016C, HHSN268200900017C, HHSN268200900018C, HHSN268200900019C, and HHSN268200900020C) and grants from the NIH/ NHLBI (grants U01 HL137880 and U24 HL141762) and supplemented through contributions made to the Foundation for the NIH and the COPD Foundation by AstraZeneca, MedImmune, Bayer Corporation (Whippany, New Jersey), Bellerophon Therapeutics (Warren, New Jersey), Boehringer-Ingelheim, Chiesi Farmaceutici S.p.A. (Parma, Italia), the Forest Research Institute, Inc. (Jersey City, New Jersey), GlaxoSmithKline, Grifols Therapeutics, Inc. (Research Triangle Park, North Carolina), Ikaria, Inc. (Hampton, New Jersey), Novartis, Nycomed Pharma GmbH (Zurich, Switzerland), ProterixBio, Inc. (Billerica, Massachusetts), Regeneron, Sanofi, Sunovion, Takeda Pharmaceutical Company (Tokyo, Japan), Theravance Biopharma, Inc. (South San Francisco, California), and Mylan N.V. (White Sulphur Springs, West Virginia). SHS The Strong Heart Study has been funded in whole or in part with federal funds from the National Heart, Lung, and Blood Institute, National Institute of Health, Department of Health and Human Services, under contract numbers 75N92019D00027, 75N92019D00028, 75N92019D00029, & 75N92019D00030. The study was previously supported by research grants: R01HL109315, R01HL109301, R01HL109284, R01HL109282, and R01HL109319 and by cooperative agreements: U01HL41642, U01HL41652, U01HL41654, U01HL65520, and U01HL65521. The content is solely the responsibility of the authors and does not necessarily represent the official views of the National Institutes of Health or the Indian Health Service (IHS). The funders had no role in study design, data collection and analysis, decision to publish, or preparation of the manuscript.

**Competing interests:** A Krishnaswamy left Columbia University and joined Merck (Epidemiology) as a project manager following completion of her work on this paper. Mitchell Elkind receives royalties from UpToDate for a chapter on neurological complications of COVID-19; receives study drug in kind from the BMS-Pfizer Alliance for Eliquis and ancillary funding from Roche, both for an NIH-funded trial of stroke prevention. E B Levitan receives research funding from Amgen (to UAB) unrelated to the current work and personal fees for serving on a DSMB

## COVID-19 events ascertainment

Potential COVID-19 hospitalizations and deaths were ascertained by cohorts via several mechanisms. C4R questionnaires, which were administered in two waves across all 14 cohorts, included questions regarding hospitalization for COVID-19 (S3 Table). If necessary (e.g., in cases of participant dementia or death), questionnaires were administered to a participant's proxy. Additional COVID-19-related hospitalizations and deaths were identified by regular non-C4R follow-up calls conducted in 9 cohorts (ARIC, CARDIA, FHS, HCHS/ SOL, JHS, MESA, NOMAS, REGARDS, SHS) to collect information on all-cause hospitalization and vital status. These data were supplemented with information collected at in-person exams, which were conducted in all cohorts during the pandemic period, except for NOMAS and REGARDS. Various non-questionnaire ascertainment methods, such as active surveillance of local EHR systems and other sources (e.g., obituaries), were performed by ARIC, FHS, JHS, SARP, SHS, and selected clinical sites in CARDIA, COPDGene, and MESA. Most cohorts supplemented vital status data with National Death Index (NDI) searches, although these are subject to reporting lags and were not available for the pandemic period at the time of this report.

After confirming participant/proxy consent, cohort staff requested copies of medical records for potential COVID-19-related hospitalizations and deaths, including physician notes (admission, consultation, discharge), radiology and laboratory reports, electrocardiogram reports, and discharge diagnoses (including discharge diagnosis ICD codes). Where applicable, death certificates were obtained. Medical records submitted for central review at the DCHC were de-identified prior to secure file transfer.

## Adjudication

**Eligibility for adjudication.** Hospitalizations (fatal and non-fatal) and out-of-hospital deaths were eligible for protocolized medical record review if they were assigned, in any position, any of the following COVID-19 discharge diagnoses based on the International Classification of Diseases, Tenth Revision (ICD-10), Clinical Modification codes [36]: confirmed COVID-19 (U07.1), post-infectious state after COVID-19 (U09.9), Multisystem inflammatory syndrome associated with COVID-19 (M35.81), Personal History of COVID-19 (Z86.16), Pneumonia due to coronavirus disease 2019 (J12.82), Other viral pneumonia (J12.89), or, for events occurring prior to May 1 2020, other coronavirus (B97.29). In the absence of these discharge diagnoses, records were considered eligible for review if there was evidence of a positive COVID-19 test, physician suspicion of COVID-19, or next-of-kin interview indicating suspected or known COVID-19 infection. Episodes of treatment in the Emergency Department (ED) for >24 hours were classified as hospitalizations since many hospitals used the ED for inpatient care during the heights of COVID-19 surges.

**Adjudication process.** Eligible COVID-19-related hospitalizations (fatal and non-fatal) and out-of-hospital deaths were subjected to adjudication by physicians and/or clinical nurse practitioners with experience in evaluating hospitalized COVID-19. Reviewers were trained via webinar and individualized instruction. Three cohorts (FHS, REGARDS, SARP) elected to perform review by local adjudicators; records from the remaining cohorts were reviewed centrally at the DCHC. Adjudication entailed data abstraction, including information on oxygenation levels and medication administration, followed by classification of COVID-19 outcomes. All COVID-19 outcomes were defined as definite, probable, or absent, based on specific criteria, including symptoms and test results, and adjudicator judgment (S4 Table).

**Adjudication of COVID-19 infection.** Adjudication of Definite COVID-19 Infection required evidence of a positive SARS-CoV-2 test. Per protocol, an administrative criterion

from University of Pittsburgh for an NIH funded trial. David Schwartz is founder and chief scientific officer of Eleven P15, a company dedicated to the diagnosis, prevention, and treatment of early presentations of pulmonary fibrosis. Joyce Lee serves as a consultant for Eleven P15. Sally Wenzel receives funding for consulting and clinical trials from AstraZeneca, GSK, Sanofi-Genzyme, Novartis, Knopp; she also receives research support from Pieris and Regeneron. PG Woodruff has served as a consultant for Roche, Sanofi, Amegne and Astra Zeneca outside the context of this paper. This does not alter our adherence to PLOS ONE policies on sharing data and materials.

(i.e., assignment of ICD U07.1) was sufficient to classify Definite COVID-19 infection, since this diagnosis code specifically requires positive testing for SARS-CoV-2; of note, for all other outcomes, administrative criteria were insufficient for definite classification but could be used to justify a probable classification. Definite COVID-19 infection was necessary to define any other COVID-19 outcome as definite. Adjudication of Probable COVID-19 Infection required anticipated signs and symptoms of COVID-19 without confirmatory testing. In the absence of adjudicated Definite or Probable COVID-19 infection, no additional outcomes were adjudicated.

**Adjudication of COVID-19 hospitalization.** In the context of Definite or Probable SARS-CoV-2 infection, hospitalization "due to" COVID-19 was defined as being admitted to the hospital for COVID-19-related signs or symptoms; developing COVID-19-related signs or symptoms during hospitalization; or, death in the ED. Whereas, hospitalization "with" COVID-19 was defined as being diagnosed with SARS-CoV-2 infection, yet being admitted to the hospital for a reason other than COVID-19 related signs or symptoms, without developing COVID-19 signs or symptoms during hospitalization.

**Adjudication of COVID-19 severity.** Definitions of severe and critical disease were based on NIH COVID-19 treatment guidelines [37].

**Adjudication of COVID-19 complications.** While treating physician notes and administrative criteria could be used for Probable classification of COVID-19 complications, Definite classification of COVID-19 pneumonia, pulmonary embolus (PE), deep vein thrombosis (DVT), or stroke required radiologic evidence in the medical record. Definite classification of acute kidney injury (AKI) was based on reported blood creatinine levels or initiation of renal replacement therapy. Definite COVID-19 myocardial infarction (MI) was defined based on biomarker and electrocardiographic or pathologic criteria to identify cases of MI caused by acute atherothrombotic coronary artery disease, or "Type 1" MI, in the context of COVID-19 infection [38]. Probable COVID-19 MI was defined more broadly and likely captures Type 1 and Type 2 MI. Myocardial injury was defined as a maximum recorded troponin level that was greater than two times the upper limit of normal (ULN), with or without associated ischemic symptoms.

**Re-adjudication.** Following completion of adjudication, reviewers could request a second independent adjudication for challenging cases. A random 10% subset of records was also submitted for a second independent adjudication.

**Analysis.** Characteristics of C4R participants, with and without an ascertained or adjudicated COVID-19 event, were tabulated. The incidence of events ascertained and adjudicated per month (based on date of event) was plotted. Since we found that records relating to out-of-hospital deaths (e.g., death certificates) did not contain sufficient data for adjudication, the following analyses were limited to adjudicated fatal and non-fatal COVID-19 hospitalizations among participants who consented to data sharing on the C4R Analysis Commons. The incidence of COVID-19-related outcomes was tabulated, by level of certainty. Interrater agreement was assessed via positive agreement, negative agreement, and the Cohen's κ-statistic [39]. The performance of discharge diagnosis ICD code-based classification was compared against definite C4R classifications, which were treated as the reference standards; probable classifications were excluded from these comparisons because they included discharge diagnosis ICD codes in the diagnostic criteria. Sensitivity, specificity, positive predictive value (PPV) and negative predictive value (NPV) were calculated. Also, myocardial injury was compared to adjudicated classifications of definite COVID-19 MI. Analyses were performed in SAS Studio software (SAS Institute, Cary, NC) on the C4R Analysis Commons.

## Results

### COVID-19 events ascertainment

Among 49,790 C4R participants, 1,974 potential COVID-19-related hospitalization and/or deaths among 1,772 (3.6%) participants were ascertained between January 2020 and June 2023 (Fig 1), for an estimated incidence density rate of 11.9 per 1,000 person-years of follow-up. Overall, the cohorts reported that half (53%) of these events were identified by a C4R questionnaire. As of June 2023, 1,768 (90%) medical records were requested, of which 1,523 (86%) were obtained. The most common reasons for failure to obtain medical records were inability to obtain participant consent and lack of response from the hospital.

Following 3 years of protocol funding (October 1, 2020—October 1, 2023), 1,237 of 1,974 (63%) events were adjudicated over 28 months. After exclusion of 57 out-of-hospital deaths due to incomplete information and 13 non-fatal hospitalizations with consent restrictions, there were 1,167 (59% of the 1,974) events available for analysis, of which 1,030 had evidence of COVID-19 (88% of 1,167 events available for analysis) (Fig 2).

Table 1 compares socio-demographic and clinical characteristics of 1,772 participants among whom a potential COVID-19 hospitalization was ascertained, including those for whom events were (N = 1098) and were not yet (N = 674) adjudicated as of October 2023,

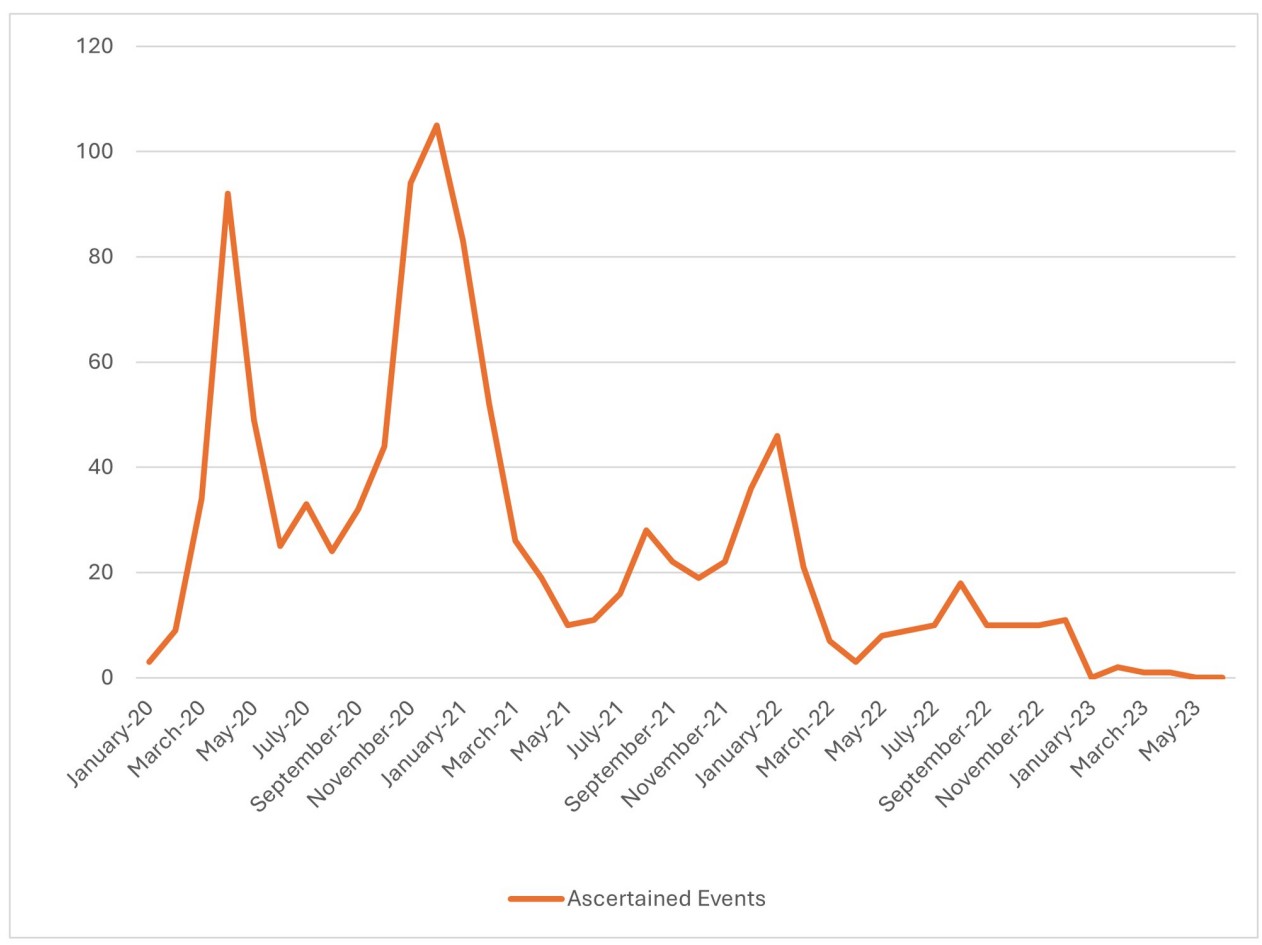

**Fig 1. Incidence of COVID-19-related hospitalization and/or death over C4R follow-up, United States, January 2020—June 2023.** Incidence is calculated per month and based on date of event.

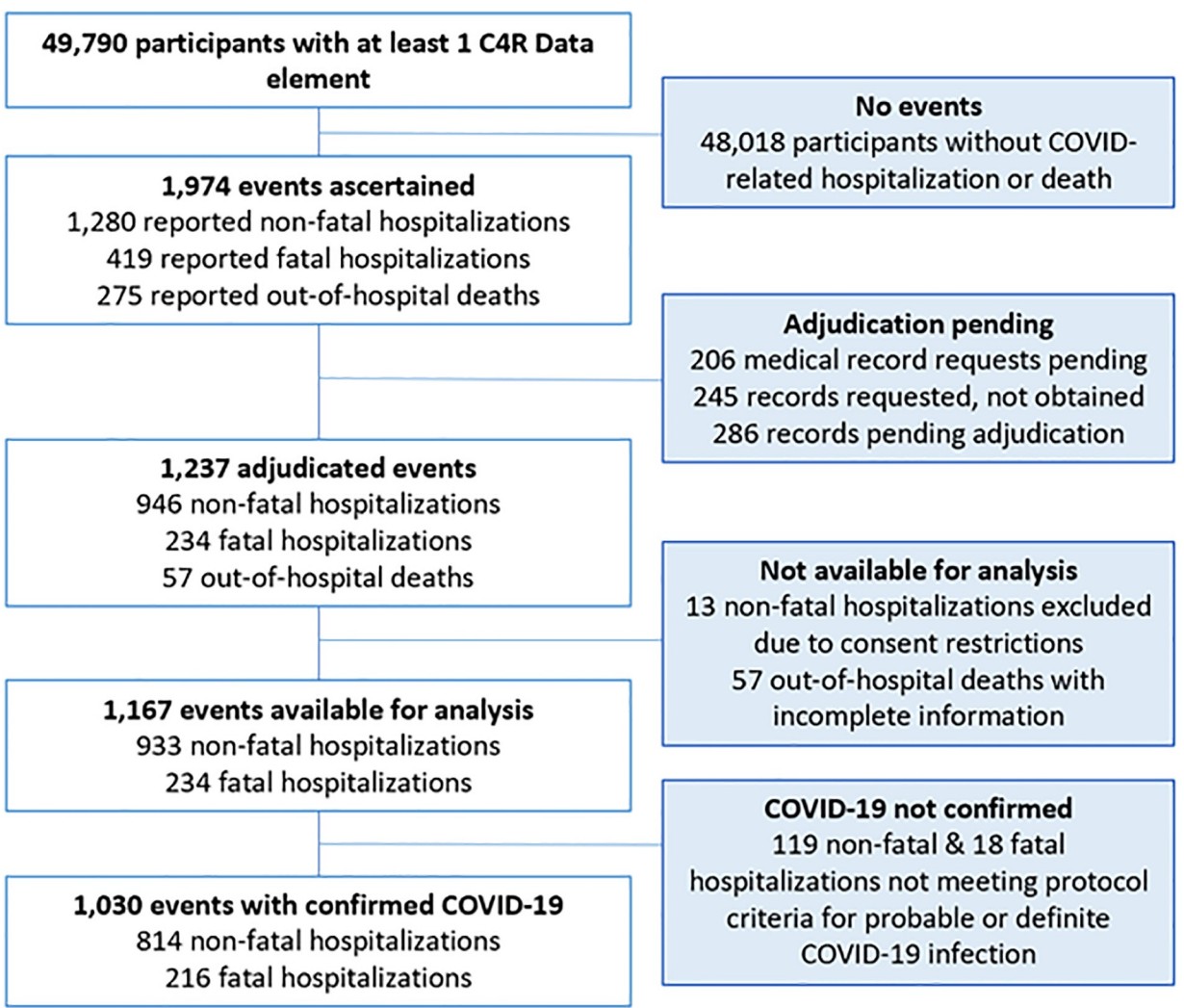

**Fig 2. Consort diagram of participants with COVID-19-related hospitalizations and deaths ascertained by C4R, January 2020—June 2023.**

versus those with no COVID-19 hospitalization or death ascertained through June 2023 (N = 48,018). Compared to participants without a potential COVID-19 hospitalization or death, those with an ascertained event were more likely to be male, to report American Indian ancestry, and to have a history of pre-pandemic smoking, diabetes, or hypertension; they were less likely to have education beyond college and less likely to be vaccinated compared to those without ascertained events. Notably, among participants with ascertained potential COVID-19 events, characteristics of those with adjudicated records were generally similar to those pending adjudication. More than one COVID-19 hospitalization was ascertained in 50 (3%) participants.

## Adjudication

**Adjudication process.** There were 7 reviewers at the C4R DCHC, 5 at FHS, 3 at REGARDS, and 2 at SARP. The median length of medical records was 59 pages (IQR: 37, 105). The average time to review was approximately 30 minutes per record.

**Table 1. Baseline characteristics of C4R participants according to ascertainment and adjudication of COVID-19 hospitalization or death.**

| Characteristic | No event as of June 30, 2023 | | Ascertained event as of June 30, 2023; pending adjudication as of October 1, 2023 | | Adjudicated event | |
|---|---|---|---|---|---|---|
| | N | % [b] | N | % [b] | N | % [b] |
| Participants[a], n (%) | 48018 | | 674 | | 1098 | |
| COVID-19 hospitalizations and deaths, n | | | 737 | | 1237 | |
| Sex | | | | | | |
| Male | 19932 | 42 | 304 | 48 | 456 | 46 |
| Female | 28019 | 58 | 326 | 52 | 543 | 54 |
| Race/Ethnicity | | | | | | |
| Asian | 1136 | 2 | 9 | 1 | 8 | 1 |
| American Indian and Alaskan Native | 1456 | 3 | 38 | 6 | 248 | 23 |
| Black | 10733 | 22 | 210 | 33 | 209 | 19 |
| White | 22347 | 47 | 174 | 27 | 457 | 43 |
| Hispanic | 12192 | 25 | 207 | 32 | 248 | 23 |
| Other | 76 | 0 | 1 | 0 | 0 | 0 |
| Age | | | | | | |
| 18–30 years | 565 | 1 | 5 | 1 | 3 | 0 |
| 31–64 years | 16673 | 35 | 167 | 28 | 182 | 21 |
| 65+ years | 29888 | 63 | 417 | 71 | 695 | 79 |
| Cohort | | | | | | |
| ARIC | 5560 | 12 | 53 | 8 | 311 | 28 |
| CARDIA | 2798 | 6 | 29 | 4 | 27 | 2 |
| COPDGene | 4109 | 9 | 33 | 5 | 49 | 4 |
| FHS | 3300 | 7 | 15 | 2 | 52 | 5 |
| HCHS/SOL | 11021 | 23 | 132 | 20 | 191 | 17 |
| JHS | 2279 | 5 | 100 | 15 | 0 | 0 |
| MASALA | 572 | 1 | 1 | 0 | 0 | 0 |
| MESA | 3268 | 7 | 189 | 28 | 47 | 4 |
| NOMAS | 909 | 2 | 23 | 3 | 67 | 6 |
| PrePF | 622 | 1 | 3 | 0 | 3 | 0 |
| REGARDS | 10126 | 21 | 20 | 3 | 179 | 16 |
| SARP | 508 | 1 | 2 | 0 | 4 | 0 |
| SPIROMICS | 1553 | 3 | 23 | 3 | 9 | 1 |
| SHS | 1393 | 3 | 51 | 8 | 159 | 14 |
| Geographic Region | | | | | | |
| Middle Atlantic | 7804 | 16 | 145 | 22 | 274 | 25 |
| Midwest | 9605 | 20 | 140 | 21 | 154 | 14 |
| New England | 3162 | 7 | 15 | 2 | 56 | 5 |
| South | 16155 | 34 | 210 | 32 | 297 | 28 |
| Southwest | 857 | 2 | 5 | 1 | 3 | 0 |
| West | 3500 | 7 | 63 | 10 | 242 | 22 |
| Smoking Status | | | | | | |
| Current | 6610 | 14 | 97 | 16 | 128 | 13 |
| Former | 17258 | 37 | 247 | 40 | 413 | 43 |
| Never | 23339 | 49 | 270 | 44 | 413 | 43 |

*(Continued)*

**Table 1.** (Continued)

| Characteristic | No event as of June 30, 2023 | | Ascertained event as of June 30, 2023; pending adjudication as of October 1, 2023 | | Adjudicated event | |
|---|---|---|---|---|---|---|
| | N | %[b] | N | %[b] | N | %[b] |
| Body mass index | | | | | | |
| <25 kg/m$^2$ | 11115 | 24 | 90 | 15 | 169 | 18 |
| 25–29.9 kg/m$^2$ | 17031 | 37 | 210 | 35 | 309 | 33 |
| 30–34.9 kg/m$^2$ | 10727 | 23 | 165 | 27 | 262 | 28 |
| >35 kg/m$^2$ | 7493 | 16 | 139 | 23 | 206 | 22 |
| Education | | | | | | |
| Less than high school | 5608 | 12 | 109 | 18 | 178 | 19 |
| High school | 10820 | 23 | 151 | 25 | 273 | 29 |
| College | 9081 | 20 | 108 | 18 | 180 | 19 |
| Beyond college | 20969 | 45 | 239 | 39 | 316 | 33 |
| Medical History | | | | | | |
| Diabetes | 9099 | 19 | 173 | 28 | 301 | 31 |
| Hypertension | 23580 | 50 | 380 | 62 | 605 | 63 |
| Cardiovascular disease | 4712 | 11 | 80 | 14 | 165 | 19 |
| Chronic obstructive pulmonary disease | 3361 | 9 | 76 | 13 | 82 | 12 |
| Asthma | 4856 | 14 | 73 | 13 | 119 | 17 |
| Received COVID-19 vaccine | 28837 | 91 | 265 | 87 | 455 | 82 |
| Infection wave during which event occurred | | | | | | |
| First wave (WT, NE) | 1133 | 2 | 155 | 23 | 173 | 16 |
| Second wave (WT, rest of US) | 573 | 1 | 100 | 15 | 136 | 12 |
| Third wave (Alpha, winter) | 1487 | 3 | 159 | 24 | 338 | 31 |
| Fourth wave (MI, spring) | 385 | 1 | 42 | 6 | 45 | 4 |
| Fifth wave (Delta) | 809 | 2 | 64 | 10 | 131 | 12 |
| Sixth wave (Omicron) | 904 | 2 | 17 | 2 | 45 | 4 |
| Unknown wave | 1093 | 2 | 87 | 13 | 191 | 17 |
| No infection confirmed | 41634 | 87 | 50 | 7 | 39 | 4 |

ARIC = Atherosclerosis Risk in Communities Study; CARDIA = Coronary Artery Risk Development in Young Adults, COPDGene = Genetic Epidemiology of Chronic Obstructive Pulmonary Disease; FHS = Framingham Heart Study; HCHS/SOL = Hispanic Community Health Study/Study of Latinos; JHS = Jackson Heart Study; MASALA = Mediators of Atherosclerosis in South Asians Living in America; MESA = Multi-Ethnic Study of Atherosclerosis; NOMAS = Northern Manhattan Study; PrePF = Preclinical Pulmonary Fibrosis; REGARDS = Reasons for Geographic and Racial Differences in Stroke; SARP = Severe Asthma Research Program; SPIROMICS = Subpopulations and Intermediate Outcome Measures in COPD Study; SHS = Strong Heart Study.

[a]Multiple events per participant were possible which is why the number of events is greater than the number of participants. There were missing data for some pre-pandemic characteristics, in which case percentages are reported based on the number of non-missing observations.

[b]Column percentages reported.

**Adjudicated outcomes.** Definite SARS-CoV-2 infection was adjudicated in 87% of 1,167 events, whereas probable infection was adjudicated in only 1%. Among 1,030 adjudicated hospitalized events in which SARS-CoV-2 infection was definite or probable, COVID-19 illness was diagnosed as the cause of 952 (93%) of the hospitalizations. Table 2 describes the

**Table 2. Adjudicated COVID-19 outcomes for COVID-19-related hospitalizations ascertained by C4R.**

| Outcomes | Eligible for Adjudication | Certainty Level of Adjudication[a] | | | | | |
|---|---|---|---|---|---|---|---|
| | | Definite | | Probable | | Definite or probable | |
| | N | N | % | N | % | N | % |
| COVID-19 infection | 1167 | 1013 | 87 | 17 | 1 | 1030 | 88 |
| Hospitalization due to COVID-19 | 1030 | 913 | 89 | 39 | 4 | 952 | 92 |
| Severe COVID-19 illness | 1030 | 726 | 70 | 65 | 6 | 791 | 77 |
| Critical COVID-19 illness | 1030 | 275 | 27 | 44 | 4 | 319 | 31 |
| Fatal COVID-19 hospitalization | 1030 | 194 | 19 | 16 | 2 | 210 | 20 |
| Pneumonia caused by COVID-19 | 1030 | 711 | 69 | 111 | 11 | 822 | 80 |
| Renal failure caused by COVID-19 | 1030 | 240 | 23 | 110 | 11 | 350 | 34 |
| PE caused by COVID-19 | 1030 | 32 | 3 | 16 | 2 | 48 | 5 |
| DVT caused by COVID-19 | 1030 | 31 | 3 | 2 | 0 | 33 | 3 |
| Stroke caused by COVID-19 | 1030 | 13 | 1 | 12 | 1 | 25 | 2 |
| Myocardial infarction caused by COVID-19[b] | 1030 | 7 | 1 | 43 | 4 | 50 | 5 |

[a]Percentages displayed are of the total number of events eligible for adjudication of the COVID-19-related diagnosis.

[b]Definite COVID-19 myocardial infarction (MI) was defined based on biomarker and electrocardiographic or pathologic criteria to identify cases of MI caused by acute atherothrombotic coronary artery disease, or "Type 1" MI, in the context of COVID-19 infection. Probable COVID-19 MI was defined more broadly and likely captures Type 1 and Type 2 MI.

adjudicated probable and definite diagnoses of SARS-CoV-2 infection, acute COVID-19 illness severity, and acute COVID-19 complications following the criteria described in S3 Table: 77% were adjudicated as severe COVID-19, 31% as critical COVID-19, 80% had adjudicated COVID-19-associated pneumonia, and 34% had adjudicated COVID-19-associated AKI. Adjudicated fatal hospitalization due to COVID-19 and other cardiopulmonary complications such as PE, DVT, and MI were less common.

**Adjudicator agreement.** Of 139/1237 (11%) hospitalizations that underwent a second adjudication, there was almost perfect agreement for adjudication of COVID-19 infection, critical illness, stroke, PE, DVT, and fatal hospitalization (Table 3). There was also strong agreement for hospitalization, severe illness, pneumonia, and renal failure, and moderate agreement for MI.

**Comparison of adjudication vs. ICD codes.** Compared to adjudicated diagnoses with a definite certainty level, discharge ICD code-based classification was sensitive and specific for pneumonia (sensitivity = 84%, specificity = 90%), but less sensitive (57–81%) for cardiovascular and renal complications (Table 4). The PPV of ICD-based classification for COVID-19-related pneumonia (J12.82, pneumonia due to coronavirus disease 2019, J12.89, other viral pneumonia), AKI (N17, Acute kidney injury), PE (ICD Code I26, PE), DVT (I82, DVT) and stroke (I63, Cerebral infarction) was excellent, ranging from 81–97%; however, the PPV for COVID-19-related MI (I21, Acute myocardial infarction) was 33%.

**Comparison of adjudicated MI vs. myocardial injury.** A substantial number of participants with adjudicated events had myocardial injury that was not adjudicated as COVID-19-related MI. Among the 1,030 hospitalized events in which SARS-CoV-2 infection was adjudicated as definite or probable, 170 (25%) demonstrated myocardial injury. Of these 170 events with myocardial injury, 25 (15%) were assigned an ICD code for MI, 5 (3%) were adjudicated as definite COVID-19-related MI, and 38 (22%) were adjudicated as probable COVID-19-related MI; the majority (75%) were not adjudicated as definite or probable COVID-19-related MI. Of note, these groups are not directly comparable with the groups described in

**Table 3. Interrater agreement for adjudication of COVID-19 outcomes in C4R.**

| Outcome | Agreement | | Disagreement[c] | K-statistic | 95% CI |
|---|---|---|---|---|---|
| | Positive [a] | Negative [b] | | | |
| COVID-19 infection | 129 | 9 | 1 | 0.94 | 0.83–1.00 |
| Hospitalization due to COVID-19 | 120 | 15 | 3 | 0.90 | 0.78–1.00 |
| Severe COVID-19 illness | 89 | 40 | 9 | 0.85 | 0.76–0.94 |
| Critical COVID-19 illness | 40 | 94 | 3 | 0.95 | 0.89–1.00 |
| Fatal COVID-19 hospitalization | 25 | 110 | 2 | 0.95 | 0.89–1.00 |
| Pneumonia caused by COVID-19 | 102 | 29 | 7 | 0.86 | 0.76–0.96 |
| Renal failure caused by COVID-19 | 45 | 88 | 6 | 0.90 | 0.83–0.98 |
| PE caused by COVID-19 | 8 | 130 | 0 | 1.00 | 1.00–1.00 |
| DVT caused by COVID-19 | 4 | 134 | 0 | 1.00 | 1.00–1.00 |
| Stroke caused by COVID-19 | 4 | 134 | 0 | 1.00 | 1.00–1.00 |
| Myocardial infarction caused by COVID-19 | 4 | 128 | 5 | 0.60 | 0.28–0.92 |

CI = confidence interval.

[a]Positive agreement was defined as both reviewers confirming the outcome with either probable or definite certainty.

[b]Negative agreement was defined as both reviewers classifying the outcome as neither probable nor definite.

[c]Disagreement was defined as one reviewer classifying the outcome as probable or definite, and the other reviewer classifying the outcome as neither probable nor definite.

**Table 4. Sensitivity, specificity, and predictive values for discharge diagnosis codes versus adjudicated definite COVID-19 outcomes.**

| Discharge diagnosis ICD code, Label | Reference standard (adjudicated outcome) | Agreement and disagreement of discharge diagnosis code with adjudicated definite outcome, N | | | | Performance of discharge diagnosis code versus adjudicated definite outcome, % | | | |
|---|---|---|---|---|---|---|---|---|---|
| | | ICD code confirmed by review | False positive by ICD code vs review | False negative by ICD code vs review | Outcome not confirmed by ICD or review | Sensitivity | Specificity | PPV | NPV |
| J12.82, Pneumonia due to coronavirus disease 2019 J12.89, Other viral pneumonia | Pneumonia caused by COVID-19 | 578 | 19 | 110 | 170 | 84% | 90% | 97% | 61% |
| N17, Acute kidney injury | Renal failure caused by COVID-19 | 146 | 23 | 88 | 617 | 62% | 96% | 86% | 88% |
| I26, Pulmonary embolism | PE caused by COVID-19 | 26 | 3 | 6 | 926 | 81% | 100% | 90% | 99% |
| I82, Deep venous thrombosis | DVT caused by COVID-19 | 21 | 5 | 9 | 937 | 70% | 99% | 81% | 99% |
| I63, Cerebral infarction | Stroke caused by COVID-19 | 8 | 0 | 5 | 952 | 62% | 100% | 100% | 99% |
| I21, Acute myocardial infarction | Myocardial infarction caused by COVID-19 | 4 | 8 | 3 | 920 | 57% | 99% | 33% | 100% |

[a]Table does not include comparisons between ICD-based classification and adjudicated diagnoses with a certainty level of 'probable.' Records without an assigned discharge diagnosis code are also excluded.

DVT = deep venous thrombosis; ICD = international classification of diseases; MI = myocardial infarction; NPV = negative predictive value; PE = pulmonary embolism; PPV = positive predictive value.

Table 4, which only included events with ICD codes available and excluded events with probable MI diagnoses.

## Discussion

Protocolized adjudication confirmed four out of five hospitalizations for COVID-19 in a US meta-cohort of prospective epidemiology cohorts and adjudicated cases of pneumonia, PE, and other conditions that were not indicated by discharge diagnosis codes. These results illustrate the importance of systematic medical record review for robust classification of COVID-19 outcomes in epidemiology cohort studies, which provide unique opportunities to study antecedent risk factors for acute COVID-19 and post-COVID conditions.

Over three years, C4R ascertained at least one potential COVID-19-related hospitalization or out-of-hospital death in 3.6% of participants, equivalent to an incidence per 1,000 person-years of follow-up of 11.9. For comparison, in 8 of the C4R cohorts, the incidence per 1,000 person-years for atherosclerotic cardiovascular disease events has been estimated at 9.7 [40]. Hence, the results of C4R events ascertainment to date highlight the major impact of COVID-19 on cohort participants and provide a substantial, longitudinal dataset to support well-powered analyses of risk factors and sequelae.

Nonetheless, our experience highlights certain limitations of standard cohort events surveillance for ascertainment of COVID-19-related events. Of hospitalizations that were ascertained as potentially COVID-19-related, SARS-CoV-2 infection could not be confirmed in one in eight events. These cases could have been due to incorrect self-reporting of a COVID-19 hospitalization on a C4R or cohort questionnaire, or active surveillance methods that were more sensitive than specific. Of note, some cases where infection could not be confirmed may have been true infections, but the medical records lacked sufficient detail for adjudication. Hence, rather than censoring all non-confirmed events, C4R is assigning a certainty level for all its COVID-19 outcomes, so that investigators can select the outcome most suitable for their specific research needs.

Our findings also demonstrate the limitations of discharge diagnosis codes to define hospitalization "for" versus "with" SARS-CoV-2 infection. Of hospitalizations with confirmed SARS-CoV-2 infection, symptoms of COVID-19 illness did not contribute to the hospitalization in 7% of cases. These findings are similar to prior reports examining the role of SARS-CoV-2 in hospitalizations using EHR systems. Adjudicated C4R outcomes will allow investigators to exclude these cases of hospitalization with incidental COVID-19 from studies designed to examine risks and sequelae of severe COVID-19 illness.

Furthermore, we found that discharge diagnosis codes were insensitive for cardiopulmonary and renal complications. We adjudicated a substantial number of cases of COVID-19 pneumonia, AKI, and MI among records without a corresponding discharge diagnosis code. This may be related, in part, to the major challenges of charting during COVID-19 surges, when documentation standards were modified to prioritize direct patient care activities. Nonetheless, before the pandemic period, previous investigations on the use of administrative data to identify cases of pneumonia have found that ICD codes are imprecise and can result in a substantial number of pneumonia cases going undetected [41, 42]. These results are supported by earlier findings that ICD-10 code N17 often misses AKI during hospitalizations for kidney transplant patients [43]. Potential misestimation of the incidence of MI using claims or administrative data has also been well-described [44–47]. This has been one justification for long-standing—albeit labor- and time-intensive—ASCVD events adjudication programs in many of the C4R cohorts.

As expected, we found that a substantial proportion of hospitalized events included evidence of myocardial injury, defined by troponin values two times greater than the upper limit of normal; however, only a small subset of these cases was adjudicated as COVID-19 MI. We elected to restrict definite MI to confirmed cases of type 1 (ST-elevation MI) or type 3 MI (MI resulting in death with pathological evidence of MI). It may be difficult to differentiate type 2 MI/supply-demand mismatch, which requires documentation of a rise and fall in troponin levels and symptoms of ischemia, from myocardial injury (at least one elevated troponin level), based solely on review of obtained medical records. Prior studies demonstrated that elevated troponin values are common in patients hospitalized with COVID-19 [48–50], and may occur with conditions other than MI, such as cardiomyopathy, acute cor pulmonale, arrhythmias, or cardiogenic shock [51]. Our results suggest that the true incidence of type 1 MI, due to plaque disruption and thrombosis leading to coronary occlusion, was rare in the context of acute COVID-19, supporting the utility of protocolized adjudication in validating these outcomes for COVID-19 cardiovascular research.

Altogether, our findings have several implications for EHR-based studies that do not adjudicate endpoints. Our results suggest that ICD-based events definitions could overestimate the number of cases of hospitalization due to COVID-19 illness and underestimate COVID-related complications. In addition to incorrectly estimating the prevalence and incidence of COVID-related outcomes, the observed measurement errors could reduce the robustness and reproducibility of epidemiologic analyses. If ICD misclassification was nondifferential according to risk factors of interest, it would reduce precision and increase the risk of type 2 error; if misclassification was associated with a risk factor of interest, this could potentially bias results away from the null hypothesis. Of note, EHR-based studies often use complex algorithms to define clinical outcomes and do not rely on ICD codes alone. Nonetheless, our findings underscore the importance of validating outcome definitions via comparison with robust approaches such as protocolized events adjudication.

## Strengths and limitations

Strengths of this study include prospective ascertainment of potential COVID-19-related hospitalizations and deaths in a well-characterized, multi-ethnic, US community-based sample of adults that is relatively free of referral, survival, and recall biases compared with studies that only include hospitalized patients. The adjudication protocol, which was modeled on gold-standard epidemiology cohort events adjudication in many of the C4R cohorts, was fully standardized across the cohorts and implemented at scale to generate robust events data for analysis within three years of program initiation.

Nonetheless, certain limitations must be considered, in addition to those noted above. Medical records have not yet been obtained for 37% of cases due to lack of participant consent, lack of response from hospital systems, or other operational delays that are common to cohort events ascertainment operations. This highlights the need for novel approaches for cohorts to access medical records, such as the consenting of participants to share their own electronic medical records—now available to patients via the 21st Century Cures Act [52]—with cohorts [52]. The characteristics of participants with missing versus available medical records in C4R were comparable, so there was no clear evidence of selection bias. Some medical records had incomplete data available for adjudication, which could be related to relaxation of documentation requirements during the pandemic. Few out-of-hospital deaths were ascertained, and medical records for out-of-hospital deaths were often incomplete, due to various reasons, including lack of court documents from the family of the deceased to obtain records, or hospital decision to not grant access to the decedent's medical records for research purposes. To

address this limitation, additional information on out-of-hospital deaths associated with COVID-19 will be obtained from the NDI, which provides complete ascertainment of deaths in the US, including ICD-codes; unfortunately, there are substantial time lags inherent in NDI reporting.

## Conclusions

Ascertainment and adjudication of COVID-19 hospitalizations and deaths in longstanding NIH-funded cohort studies were feasible, albeit time- and resource-intensive, and our results illustrate the importance of systematic medical records adjudication for robust classification of COVID-19 events. Adjudication confirmed SARS-CoV-2 infection in 88% of ascertained events and found that infection may have been incidental to the hospitalization in 7% of the cases. Compared to adjudication, discharge diagnosis codes were insensitive for acute cardio-vascular and renal complications of COVID-19. Novel approaches to expedite medical records access and linkage would augment unique opportunities for COVID-19 and other health out-comes research, particularly for emerging diseases, in NIH-funded epidemiology cohorts.

## Supporting information

**S1 Appendix. Cohort descriptions.**
(DOCX)

**S2 Appendix. C4R events protocol.**
(PDF)

**S1 Table. Characteristics of participants in C4R cohorts, United States, March 1, 2020.**
(DOCX)

**S2 Table. Cohort Institutional Review Boards (IRBs) supervising implementation of the C4R protocols.**
(DOCX)

**S3 Table. Selected COVID-19 questionnaire elements and their inclusion by cohorts in questionnaires for C4R, United States, April 2020–February 2023.**
(DOCX)

**S4 Table. Criteria for classification of COVID-19 diagnoses as definite or probable based on medical record review.**
(DOCX)

## Acknowledgments

We thank the participants of each cohort for their dedication to the studies. The authors thank the staff and participants of the ARIC study for their important contributions.

The authors thank the staff and participants of the CARDIA study for their important con-tributions. Framingham Heart Study wished to acknowledge the study participants and staff members who supported the FHS-C4R effort: Cathy D'Augustine, Lindsay Clayson, Karen Mutalik, and Ken Nieto. We appreciate the support and guidance of investigators from the Study Design and Molecular Epidemiology Core of the Vermont Center for Cardiovascular and Brain Health. The authors thank the other investigators, the staff, and the participants of the REGARDS study for their valuable contributions. In particular, the REGARDS would like to thank Karen Marshall, Melissa Garner, J. David Rhodes, April Sisson, Todd M. Brown, and Parag Goyal. A full list of participating REGARDS investigators and institutions can be found

at: https://www.uab.edu/soph/regardsstudy/. The authors thank the UNC CSCC, including Lisa Reeves and Kwanhye Jung for their many contributions. The authors express gratitude to the reviewers and abstractors involved in the morbidity and mortality surveillance, as well as to the other study investigators, the dedicated study staff, and the participants of the Strong Heart Study for their significant contributions. The authors thank Milagros Ventura and Greg Neils of the Columbia Data Coordinating Center for management of secure data transmission and development of the REDCap system to capture adjudication data. The authors thank Consuelo Mora McLaughlin and Janet T De Rosa from Columbia University for their work on NOMAS events ascertainment.

## Author Contributions

**Conceptualization:** Elizabeth C. Oelsner, Tauqeer Ali, Amber Pirzada, Wendy S. Post.

**Data curation:** Elizabeth C. Oelsner, Akshaya Krishnaswamy, Rafail Rustamov, Pallavi P. Balte, Tauqeer Ali, Howard F. Andrews, Pramod Anugu, Alexander Arynchyn, Lori A. Bateman, Jianwen Cai, Harry Chang, Lucas Chen, Karen Hinckley Stukovsky, Lauren Jager, Ling Jin, Suzanne E. Judd, Maureen R. Kelly, Emily B. Levitan, Kimberly Malloy, David Mauger, Charles G. Murphy, Ashmi A. Patel, Kimberly B. Ring, James M. Shikany, Daniela Sotres-Alvarez, Cheryl Tarlton, Maya Vankineni, Sally E. Wenzel, Ji Hyun Yang.

**Formal analysis:** Akshaya Krishnaswamy, Rafail Rustamov, Pallavi P. Balte.

**Funding acquisition:** Elizabeth C. Oelsner, Tauqeer Ali, Norrina B. Allen, Howard F. Andrews, Jianwen Cai, Mitchell S. V. Elkind, Kelley Pettee Gabriel, Jose D. Gutierrez, Virginia J. Howard, Carmen R. Isasi, Suzanne E. Judd, Namratha R. Kandula, Barry J. Make, Jennifer J. Manly, Joanne M. Murabito, Victor E. Ortega, Elizabeth A. Regan, David A. Schwartz, James M. Shikany, Sally E. Wenzel, Ying Zhang, Wendy S. Post.

**Investigation:** Elizabeth C. Oelsner, Rafail Rustamov, Tauqeer Ali, Norrina B. Allen, Howard F. Andrews, Alexander Arynchyn, Jianwen Cai, Harry Chang, Lucas Chen, Mitchell S. V. Elkind, James S. Floyd, Kelley Pettee Gabriel, Sina A. Gharib, Jose D. Gutierrez, Virginia J. Howard, Carmen R. Isasi, Ling Jin, Suzanne E. Judd, Alka M. Kanaya, Namratha R. Kandula, Maureen R. Kelly, Sadiya S. Khan, Anna Kucharska-Newton, Joyce S. Lee, Emily B. Levitan, Cora E. Lewis, Barry J. Make, Jennifer J. Manly, Yuan-I Min, Joanne M. Murabito, Charles G. Murphy, Arnita F. Norwood, George T. O'Connor, Victor E. Ortega, Ashmi A. Patel, Amber Pirzada, Elizabeth A. Regan, Wayne D. Rosamond, David A. Schwartz, James M. Shikany, Daniela Sotres-Alvarez, Janis Tse, Elman M. Urbina Meneses, Maya Vankineni, Sally E. Wenzel, Prescott G. Woodruff, Vanessa Xanthakis, Ji Hyun Yang, Neil A. Zakai, Ying Zhang, Wendy S. Post.

**Methodology:** Elizabeth C. Oelsner, Tauqeer Ali, Norrina B. Allen, Howard F. Andrews, James S. Floyd, Sina A. Gharib, Karen Hinckley Stukovsky, Virginia J. Howard, Carmen R. Isasi, Suzanne E. Judd, Alka M. Kanaya, Namratha R. Kandula, Sadiya S. Khan, Anna Kucharska-Newton, Joyce S. Lee, Emily B. Levitan, Cora E. Lewis, Barry J. Make, Jennifer J. Manly, Yuan-I Min, Joanne M. Murabito, Arnita F. Norwood, George T. O'Connor, Victor E. Ortega, Elizabeth A. Regan, Kimberly B. Ring, Wayne D. Rosamond, David A. Schwartz, James M. Shikany, Daniela Sotres-Alvarez, Janis Tse, Sally E. Wenzel, Prescott G. Woodruff, Vanessa Xanthakis, Neil A. Zakai, Ying Zhang, Wendy S. Post.

**Project administration:** Akshaya Krishnaswamy, Rafail Rustamov, Pallavi P. Balte, Tauqeer Ali, Norrina B. Allen, Howard F. Andrews, Pramod Anugu, Alexander Arynchyn, Lori A. Bateman, Jianwen Cai, Mitchell S. V. Elkind, Kelley Pettee Gabriel, Jose D. Gutierrez,

Karen Hinckley Stukovsky, Virginia J. Howard, Carmen R. Isasi, Lauren Jager, Suzanne E. Judd, Alka M. Kanaya, Namratha R. Kandula, Anna Kucharska-Newton, Joyce S. Lee, Emily B. Levitan, Cora E. Lewis, Barry J. Make, Kimberly Malloy, David Mauger, Yuan-I Min, Joanne M. Murabito, Arnita F. Norwood, Victor E. Ortega, Amber Pirzada, Elizabeth A. Regan, Kimberly B. Ring, Wayne D. Rosamond, David A. Schwartz, James M. Shikany, Daniela Sotres-Alvarez, Cheryl Tarlton, Janis Tse, Elman M. Urbina Meneses, Sally E. Wenzel, Prescott G. Woodruff, Vanessa Xanthakis, Ying Zhang, Wendy S. Post.

**Resources:** Elizabeth C. Oelsner, Tauqeer Ali, Norrina B. Allen, Howard F. Andrews, Pramod Anugu, Alexander Arynchyn, Jianwen Cai, Mitchell S. V. Elkind, Kelley Pettee Gabriel, Jose D. Gutierrez, Karen Hinckley Stukovsky, Virginia J. Howard, Carmen R. Isasi, Lauren Jager, Suzanne E. Judd, Alka M. Kanaya, Namratha R. Kandula, Joyce S. Lee, Emily B. Levitan, Cora E. Lewis, Barry J. Make, Kimberly Malloy, David Mauger, Yuan-I Min, Joanne M. Murabito, Arnita F. Norwood, Victor E. Ortega, Elizabeth A. Regan, Kimberly B. Ring, David A. Schwartz, James M. Shikany, Daniela Sotres-Alvarez, Cheryl Tarlton, Janis Tse, Sally E. Wenzel, Prescott G. Woodruff, Vanessa Xanthakis, Ying Zhang, Wendy S. Post.

**Software:** Howard F. Andrews, Alexander Arynchyn, Suzanne E. Judd, Cora E. Lewis, Kimberly B. Ring, James M. Shikany, Janis Tse.

**Supervision:** Elizabeth C. Oelsner, Tauqeer Ali, Suzanne E. Judd, Emily B. Levitan, Janis Tse, Wendy S. Post.

**Validation:** Pallavi P. Balte, Janis Tse.

**Writing – original draft:** Elizabeth C. Oelsner, Akshaya Krishnaswamy.

**Writing – review & editing:** Elizabeth C. Oelsner, Rafail Rustamov, Pallavi P. Balte, Tauqeer Ali, Norrina B. Allen, Howard F. Andrews, Pramod Anugu, Alexander Arynchyn, Lori A. Bateman, Jianwen Cai, Harry Chang, Lucas Chen, Mitchell S. V. Elkind, James S. Floyd, Kelley Pettee Gabriel, Sina A. Gharib, Jose D. Gutierrez, Karen Hinckley Stukovsky, Virginia J. Howard, Carmen R. Isasi, Lauren Jager, Ling Jin, Suzanne E. Judd, Alka M. Kanaya, Namratha R. Kandula, Maureen R. Kelly, Sadiya S. Khan, Anna Kucharska-Newton, Joyce S. Lee, Emily B. Levitan, Cora E. Lewis, Barry J. Make, Kimberly Malloy, Jennifer J. Manly, David Mauger, Yuan-I Min, Joanne M. Murabito, Charles G. Murphy, Arnita F. Norwood, George T. O'Connor, Victor E. Ortega, Ashmi A. Patel, Amber Pirzada, Elizabeth A. Regan, Kimberly B. Ring, Wayne D. Rosamond, David A. Schwartz, James M. Shikany, Daniela Sotres-Alvarez, Cheryl Tarlton, Janis Tse, Elman M. Urbina Meneses, Maya Vankineni, Sally E. Wenzel, Prescott G. Woodruff, Vanessa Xanthakis, Ji Hyun Yang, Neil A. Zakai, Ying Zhang, Wendy S. Post.

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
