## [Decision Letter · Decision Letter 0]

4 Nov 2024

PONE-D-24-32613

Classifying COVID-19 Hospitalizations in Epidemiology Cohort Studies: The C4R Study

PLOS ONE

Dear Dr. Oelsner,

Thank you for submitting your manuscript to PLOS ONE. After careful consideration, we feel that it has merit but does not fully meet PLOS ONE’s publication criteria as it currently stands. Therefore, we invite you to submit a revised version of the manuscript that addresses the points raised during the review process.

We look forward to receiving your revised manuscript.

Kind regards,

Kamal Sharma

Academic Editor

PLOS ONE

Journal Requirements: When submitting your revision, we need you to address these additional requirements. 1. Please ensure that your manuscript meets PLOS ONE's style requirements, including those for file naming. The PLOS ONE style templates can be found at https://journals.plos.org/plosone/s/file?id=wjVg/PLOSOne_formatting_sample_main_body.pdf and https://journals.plos.org/plosone/s/file?id=ba62/PLOSOne_formatting_sample_title_authors_affiliations.pdf 2. We note that the grant information you provided in the ‘Funding Information’ and ‘Financial Disclosure’ sections do not match.  When you resubmit, please ensure that you provide the correct grant numbers for the awards you received for your study in the ‘Funding Information’ section. 3. Thank you for stating the following financial disclosure: "The Collaborative Cohort of Cohorts for COVID-19 Research (C4R) Study is supported by grant OT2HL156812 from the NHLBI Collaborating Network of Networks for Evaluating COVID-19 and Therapeutic Strategies/Researching COVID to Enhance Recovery, with co-funding from the National Institute of Neurological Disorders and Stroke (NINDS) and the National Institute on Aging (NIA). The Principal Investigator is ECO. Additional funding for participating cohorts is as follows: ARICThe Atherosclerosis Risk in Communities Study has been funded in whole or in part by the NHLBI, National Institutes of Health (NIH), US Department of Health and Human Services, under contracts 75N92022D00001, 75N92022D00002, 75N92022D00003, 75N92022D00004, and 75N92022D00005. Neurocognitive data are collected under grants U01 2U01HL096812, 2U01HL096814, 2U01HL096899, 2U01HL096902, and 2U01HL096917 from the NHLBI, the NINDS, the NIA, and the National Institute on Deafness and Other Communication Disorders. CARDIAThe Coronary Artery Risk Development in Young Adults (CARDIA) Study is conducted and supported by the NHLBI in collaboration with the University of Alabama at Birmingham (contracts HHSN268201800005I and HHSN268201800007I), Northwestern University (contract HHSN268201800003I), the University of Minnesota (contract HHSN268201800006I), and the Kaiser Foundation Research Institute (contract HSN268201800004I). COPDGeneThe Genetic Epidemiology of COPD (COPDGene) Study was supported by awards U01 HL089897 and U01 HL089856 from the NHLBI. COPDGene is also supported by the COPD Foundation through contributions made to an industry advisory board comprised of AstraZeneca AB (Cambridge, United Kingdom), Boehringer-Ingelheim (Ingelheim am Rhein, Germany), Genentech, Inc. (South San Francisco, California), GlaxoSmithKline plc (London, United Kingdom), Novartis International AG (Basel, Switzerland), Pfizer, Inc. (New York, New York), Siemens AG (Berlin, Germany), and Sunovion Pharmaceuticals Inc. (Marlborough, Massachusetts). FHSThe Framingham Heart Study has received support from the NHLBI (grant N01-HC-25195, contract HHSN268201500001I, and grant 75N92019D00031). HCHS/SOLThe Hispanic Community Health Study/Study of Latinos (HCHS/SOL) is a collaborative study supported by contracts between the NHLBI and the University of North Carolina (contract HHSN268201300001I/N01-HC-65233), the University of Miami (contract HHSN268201300004I/N01-HC-65234), Albert Einstein College of Medicine (contract HHSN268201300002I/N01-HC-65235), the University of Illinois at Chicago (contract HHSN268201300003I/N01-HC-65236 (Northwestern University)), and San Diego State University (contract HHSN268201300005I/N01-HC-65237). The following institutes/centers/offices have contributed to the HCHS/SOL through a transfer of funds to the NHLBI: the National Institute on Minority Health and Health Disparities, the National Institute on Deafness and Other Communication Disorders, the National Institute of Dental and Craniofacial Research, the National Institute of Diabetes and Digestive and Kidney Diseases, the NINDS, and the NIH Office of Dietary Supplements. JHSThe Jackson Heart Study is supported by and conducted in collaboration with Jackson State University (contract HHSN268201800013I), Tougaloo College (contract HHSN268201800014I), the Mississippi State Department of Health (contract HHSN268201800015I), the University of Mississippi Medical Center (contracts HHSN268201800010I, HHSN268201800011I, and HHSN268201800012I), the NHLBI, and the National Institute on Minority Health and Health Disparities. MASALAThe Mediators of Atherosclerosis in South Asians Living in America (MASALA) Study was supported by grant R01HL093009 from the NHLBI, the National Center for Research Resources, and the National Center for Advancing Translational Sciences, NIH, through University of California, San Francisco–Clinical and Translational Science Institute grant UL1RR024131. MESAThe Multi-Ethnic Study of Atherosclerosis (MESA) and the MESA SNP Health Association Resource (SHARe) are conducted and supported by the NHLBI in collaboration with the MESA investigators. Support for MESA is provided by grants and contracts 75N92020D00001, HHSN268201500003I, N01-HC-95159, 75N92020D00005, N01-HC-95160, 75N92020D00002, N01-HC-95161, 75N92020D00003, N01-HC-95162, 75N92020D00006, N01-HC-95163, 75N92020D00004, N01-HC-95164, 75N92020D00007, N01-HC-95165, N01-HC-95166, N01-HC-95167, N01-HC-95168, N01-HC-95169, R01-HL077612, R01-HL093081, R01-HL130506, R01-HL127028, R01-HL127659, R01-HL098433, R01-HL101250, and R01-HL135009 from the NHLBI; grant R01-AG058969 from the NIA; and grants UL1-TR-000040, UL1-TR-001079, and UL1-TR-001420 from the National Center for Advancing Translational Sciences. NHLBI Pooled Cohorts StudyThe NHLBI Pooled Cohorts Study was supported by grants R21HL153700, K23HL130627, R21HL129924, and R21HL121457 from the NIH/NHLBI. NOMASThe Northern Manhattan Study was supported by grants R01 NS29993 and R01 NS48134 from the NINDS and grant R01 AG066162 from the NIA. PrePFThe Prevent Pulmonary Fibrosis cohort study was established in 2000 and has been supported by NIH awards Z01-ES101947, R01-HL095393, RC2-HL1011715, R21/33-HL120770, R01-HL097163, Z01-HL134585, UH2/3-HL123442, P01-HL092870, UG3/UH3-HL151865, and DoD W81XWH-17-1-0597. REGARDSThe REasons for Geographic and Racial Differences in Stroke (REGARDS) Study is supported by cooperative agreement U01 NS041588, co-funded by the NINDS and the NIA. SARPResearch by the principal and co–principal investigators of the Severe Asthma Research Program was funded by the NIH/NHLBI (grants U10 HL109164, U10 HL109257, U10 HL109146, U10 HL109172, U10 HL109250, U10 HL109168, U10 HL109152, and U10 HL109086). Additional support was provided through industry partnerships with the following companies: AstraZeneca, Boehringer-Ingelheim, Genentech, GlaxoSmithKline, MedImmune, Inc. (Gaithersburg, Maryland), Novartis, Regeneron Pharmaceuticals, Inc. (Tarrytown, New York), Sanofi S.A. (Paris, France), and Teva Pharmaceuticals USA (North Wales, Pennsylvania). Spirometers used in Severe Asthma Research Program III were provided by nSpire Health, Inc. (Longmont, Colorado). SPIROMICSThe Subpopulations and Intermediate Outcome Measures in COPD Study (SPIROMICS) has been funded by contracts with the NIH/NHLBI (contracts HHSN268200900013C, HHSN268200900014C, HHSN268200900015C, HHSN268200900016C, HHSN268200900017C, HHSN268200900018C, HHSN268200900019C, and HHSN268200900020C) and grants from the NIH/NHLBI (grants U01 HL137880 and U24 HL141762) and supplemented through contributions made to the Foundation for the NIH and the COPD Foundation by AstraZeneca, MedImmune, Bayer Corporation (Whippany, New Jersey), Bellerophon Therapeutics (Warren, New Jersey), Boehringer-Ingelheim, Chiesi Farmaceutici S.p.A. (Parma, Italia), the Forest Research Institute, Inc. (Jersey City, New Jersey), GlaxoSmithKline, Grifols Therapeutics, Inc. (Research Triangle Park, North Carolina), Ikaria, Inc. (Hampton, New Jersey), Novartis, Nycomed Pharma GmbH (Zurich, Switzerland), ProterixBio, Inc. (Billerica, Massachusetts), Regeneron, Sanofi, Sunovion, Takeda Pharmaceutical Company (Tokyo, Japan), Theravance Biopharma, Inc. (South San Francisco, California), and Mylan N.V. (White Sulphur Springs, West Virginia). SHSThe Strong Heart Study has been funded in whole or in part with federal funds from the National Heart, Lung, and Blood Institute, National Institute of Health, Department of Health and Human Services, under contract numbers 75N92019D00027, 75N92019D00028, 75N92019D00029, & 75N92019D00030.  The study was previously supported by research grants: R01HL109315, R01HL109301, R01HL109284, R01HL109282, and R01HL109319 and by cooperative agreements: U01HL41642, U01HL41652, U01HL41654, U01HL65520, and U01HL65521. The content is solely the responsibility of the authors and does not necessarily represent the official views of the National Institutes of Health or the Indian Health Service (IHS)." Please state what role the funders took in the study.  If the funders had no role, please state: ""The funders had no role in study design, data collection and analysis, decision to publish, or preparation of the manuscript."" If this statement is not correct you must amend it as needed. Please include this amended Role of Funder statement in your cover letter; we will change the online submission form on your behalf. 4. Thank you for stating the following in the Competing Interests section: "A Krishnaswamy left Columbia University and joined Merck (Epidemiology) as a project manager following completion of her work on this paper.Mitchell Elkind receives royalties from UpToDate for a chapter on neurological complications of COVID-19; receives study drug in kind from the BMS-Pfizer Alliance for Eliquis and ancillary funding from Roche, both for an NIH-funded trial of stroke prevention. E B Levitan receives research funding from Amgen (to UAB) unrelated to the current work and personal fees for serving on a DSMB from University of Pittsburgh for an NIH funded trial.David Schwartz is founder and chief scientific officer of Eleven P15, a company dedicated to the diagnosis, prevention, and treatment of early presentations of pulmonary fibrosis.  Joyce Lee serves as a consultant for Eleven P15.Sally Wenzel receives funding for consulting and clinical trials from AstraZeneca, GSK, Sanofi-Genzyme, Novartis, Knopp; she also receives research support from Pieris and Regeneron.PG Woodruff has served as a consultant for Roche, Sanofi, Amegne and Astra Zeneca outside the context of this paper." Please confirm that this does not alter your adherence to all PLOS ONE policies on sharing data and materials, by including the following statement: ""This does not alter our adherence to  PLOS ONE policies on sharing data and materials.” (as detailed online in our guide for authors http://journals.plos.org/plosone/s/competing-interests).  If there are restrictions on sharing of data and/or materials, please state these. Please note that we cannot proceed with consideration of your article until this information has been declared.  Please include your updated Competing Interests statement in your cover letter; we will change the online submission form on your behalf. 5. When completing the data availability statement of the submission form, you indicated that you will make your data available on acceptance. We strongly recommend all authors decide on a data sharing plan before acceptance, as the process can be lengthy and hold up publication timelines. Please note that, though access restrictions are acceptable now, your entire data will need to be made freely accessible if your manuscript is accepted for publication. This policy applies to all data except where public deposition would breach compliance with the protocol approved by your research ethics board. If you are unable to adhere to our open data policy, please kindly revise your statement to explain your reasoning and we will seek the editor's input on an exemption. Please be assured that, once you have provided your new statement, the assessment of your exemption will not hold up the peer review process. 6. Your ethics statement should only appear in the Methods section of your manuscript. If your ethics statement is written in any section besides the Methods, please move it to the Methods section and delete it from any other section. Please ensure that your ethics statement is included in your manuscript, as the ethics statement entered into the online submission form will not be published alongside your manuscript. 7. Please include a copy of Table 4 which you refer to in your text on page 27. 8. We note you have included a table to which you do not refer in the text of your manuscript. Please ensure that you refer to Table 5 in your text; if accepted, production will need this reference to link the reader to the Table. 9. Please include captions for your Supporting Information files at the end of your manuscript, and update any in-text citations to match accordingly. Please see our Supporting Information guidelines for more information: http://journals.plos.org/plosone/s/supporting-information. 10. Please review your reference list to ensure that it is complete and correct. If you have cited papers that have been retracted, please include the rationale for doing so in the manuscript text, or remove these references and replace them with relevant current references. Any changes to the reference list should be mentioned in the rebuttal letter that accompanies your revised manuscript. If you need to cite a retracted article, indicate the article’s retracted status in the References list and also include a citation and full reference for the retraction notice.

**Additional Editor Comments:**

Hello,

Thanks for submitting the manuscript to this journal. It is a well-written draft worthy of further evaluation.

Please address the comments raised by our reviewer for further evaluation

Thanks

Reviewers' comments:

Reviewer's Responses to Questions

**Comments to the Author**

1. Is the manuscript technically sound, and do the data support the conclusions?

Reviewer #1: Yes

2. Has the statistical analysis been performed appropriately and rigorously? 

Reviewer #1: Yes

3. Have the authors made all data underlying the findings in their manuscript fully available?

Reviewer #1: No

4. Is the manuscript presented in an intelligible fashion and written in standard English?

Reviewer #1: Yes

5. Review Comments to the Author

Reviewer #1: This manuscript by Dr Oelsner et al, “Classifying COVID-19 Hospitalizations in Epidemiology Cohort Studies: The C4R Study,” reports findings a study which used a protocolized adjudication process to compare acute COVID-19 event ascertainment by ICD code compared to manual adjudication for participants in 14 participating prospective cohort studies. They found that pneumonia and acute kidney injury were common in this population with a high proportion of critical illness, while cardiovascular and thrombotic complications were rare. There was a range of sensitivities for the various specific outcomes, suggesting that adjudication is helpful (or even necessary for accurate ascertainment of MI in particular). The research was well-done and the paper was well written, and I think this is highly worthwhile publication.

My main suggestion should the authors be asked for further revision is to add a paragraph in the discussion regarding the implication that EHR-based studies that do not include adjudication are likely to be reporting inaccurate estimates of these complications.

Other Minor Suggestions/Clarifications

Line 389: I was a bit confused about the denominator for this percentage (52% of potential COVID-19 events); should the denominator be 1,167 events available for analysis rather than the events ascertained?

Line 441, Table 2: Please put a footnote explaining how definite and probable MI are defined differently for those who miss that detail in the methods.

Page 30 (Comparison of adjudicated MI vs. myocardial injury): I was confused by the difference in numbers presented in Table 5 for MI (4 true positive, 4 false positive, 3 false negatives, and 920 true negatives) and the numbers presented in this paragraph. I think this may have happened because the table uses “Definite” but this paragraph uses “Definite or Probable.” A sentence clarifying this would be helpful.

6. PLOS authors have the option to publish the peer review history of their article (what does this mean?). If published, this will include your full peer review and any attached files.

Reviewer #1: **Yes: **Matthew S Durstenfeld, MD MAS FACC, University of California San Francisco

---

## [Author Response · Author response to Decision Letter 0]

12 Nov 2024

Response to reviewers

PONE-D-24-32613

Classifying COVID-19 Hospitalizations in Epidemiology Cohort Studies: The C4R Study

PLOS ONE

Thank you for the opportunity to revise and resubmit our manuscript. We hope you find that the revised manuscript is suitable for publication in PLOS One. Our point-by-point response is as follows: 

Response: We have revised the formatting according to the PLOS ONE style templates (https://journals.plos.org/plosone/s/file?id=wjVg/PLOSOne_formatting_sample_main_body.pdf and https://journals.plos.org/plosone/s/file?id=ba62/PLOSOne_formatting_sample_title_authors_affiliations.pdf)

Response: the correct funding statement is as follows:

C4R

The Collaborative Cohort of Cohorts for COVID-19 Research (C4R) Study is supported by grant OT2HL156812 from the NHLBI Collaborating Network of Networks for Evaluating COVID-19 and Therapeutic Strategies/Researching COVID to Enhance Recovery, with co-funding from the National Institute of Neurological Disorders and Stroke (NINDS) and the National Institute on Aging (NIA). The Principal Investigator is ECO.

 Additional funding for participating cohorts is as follows:

ARIC

The Atherosclerosis Risk in Communities Study has been funded in whole or in part by the NHLBI, National Institutes of Health (NIH), US Department of Health and Human Services, under contracts 75N92022D00001, 75N92022D00002, 75N92022D00003, 75N92022D00004, and 75N92022D00005. 

Neurocognitive data are collected under grants U01 2U01HL096812, 2U01HL096814, 2U01HL096899, 2U01HL096902, and 2U01HL096917 from the NHLBI, the NINDS, the NIA, and the National Institute on Deafness and Other Communication Disorders. 

CARDIA

The Coronary Artery Risk Development in Young Adults (CARDIA) Study is conducted and supported by the NHLBI in collaboration with the University of Alabama at Birmingham (contracts HHSN268201800005I and HHSN268201800007I), Northwestern University (contract HHSN268201800003I), the University of Minnesota (contract HHSN268201800006I), and the Kaiser Foundation Research Institute (contract HSN268201800004I). 

COPDGene

The Genetic Epidemiology of COPD (COPDGene) Study was supported by awards U01 HL089897 and U01 HL089856 from the NHLBI. COPDGene is also supported by the COPD Foundation through contributions made to an industry advisory board comprised of AstraZeneca AB (Cambridge, United Kingdom), Boehringer-Ingelheim (Ingelheim am Rhein, Germany), Genentech, Inc. (South San Francisco, California), GlaxoSmithKline plc (London, United Kingdom), Novartis International AG (Basel, Switzerland), Pfizer, Inc. (New York, New York), Siemens AG (Berlin, Germany), and Sunovion Pharmaceuticals Inc. (Marlborough, Massachusetts). 

FHS

The Framingham Heart Study has received support from the NHLBI (grant N01-HC-25195, contract HHSN268201500001I, and grant 75N92019D00031). 

HCHS/SOL

The Hispanic Community Health Study/Study of Latinos (HCHS/SOL) is a collaborative study supported by contracts between the NHLBI and the University of North Carolina (contract HHSN268201300001I/N01-HC-65233), the University of Miami (contract HHSN268201300004I/N01-HC-65234), Albert Einstein College of Medicine (contract HHSN268201300002I/N01-HC-65235), the University of Illinois at Chicago (contract HHSN268201300003I/N01-HC-65236 (Northwestern University)), and San Diego State University (contract HHSN268201300005I/N01-HC-65237). The following institutes/centers/offices have contributed to the HCHS/SOL through a transfer of funds to the NHLBI: the National Institute on Minority Health and Health Disparities, the National Institute on Deafness and Other Communication Disorders, the National Institute of Dental and Craniofacial Research, the National Institute of Diabetes and Digestive and Kidney Diseases, the NINDS, and the NIH Office of Dietary Supplements. 

JHS

The Jackson Heart Study is supported by and conducted in collaboration with Jackson State University (contract HHSN268201800013I), Tougaloo College (contract HHSN268201800014I), the Mississippi State Department of Health (contract HHSN268201800015I), the University of Mississippi Medical Center (contracts HHSN268201800010I, HHSN268201800011I, and HHSN268201800012I), the NHLBI, and the National Institute on Minority Health and Health Disparities. 

MASALA

The Mediators of Atherosclerosis in South Asians Living in America (MASALA) Study was supported by grant R01HL093009 from the NHLBI, the National Center for Research Resources, and the National Center for Advancing Translational Sciences, NIH, through University of California, San Francisco–Clinical and Translational Science Institute grant UL1RR024131. 

MESA

The Multi-Ethnic Study of Atherosclerosis (MESA) and the MESA SNP Health Association Resource (SHARe) are conducted and supported by the NHLBI in collaboration with the MESA investigators. Support for MESA is provided by grants and contracts 75N92020D00001, HHSN268201500003I, N01-HC-95159, 75N92020D00005, N01-HC-95160, 75N92020D00002, N01-HC-95161, 75N92020D00003, N01-HC-95162, 75N92020D00006, N01-HC-95163, 75N92020D00004, N01-HC-95164, 75N92020D00007, N01-HC-95165, N01-HC-95166, N01-HC-95167, N01-HC-95168, N01-HC-95169, R01-HL077612, R01-HL093081, R01-HL130506, R01-HL127028, R01-HL127659, R01-HL098433, R01-HL101250, and R01-HL135009 from the NHLBI; grant R01-AG058969 from the NIA; and grants UL1-TR-000040, UL1-TR-001079, and UL1-TR-001420 from the National Center for Advancing Translational Sciences. 

NHLBI Pooled Cohorts Study

The NHLBI Pooled Cohorts Study was supported by grants R21HL153700, K23HL130627, R21HL129924, and R21HL121457 from the NIH/NHLBI. 

NOMAS

The Northern Manhattan Study was supported by grants R01 NS29993 and R01 NS48134 from the NINDS and grant R01 AG066162 from the NIA. 

PrePF

The Prevent Pulmonary Fibrosis cohort study was established in 2000 and has been supported by NIH awards Z01-ES101947, R01-HL095393, RC2-HL1011715, R21/33-HL120770, R01-HL097163, Z01-HL134585, UH2/3-HL123442, P01-HL092870, UG3/UH3-HL151865, and DoD W81XWH-17-1-0597. 

REGARDS

The REasons for Geographic and Racial Differences in Stroke (REGARDS) Study is supported by cooperative agreement U01 NS041588, co-funded by the NINDS and the NIA. 

SARP

Research by the principal and co–principal investigators of the Severe Asthma Research Program was funded by the NIH/NHLBI (grants U10 HL109164, U10 HL109257, U10 HL109146, U10 HL109172, U10 HL109250, U10 HL109168, U10 HL109152, and U10 HL109086). Additional support was provided through industry partnerships with the following companies: AstraZeneca, Boehringer-Ingelheim, Genentech, GlaxoSmithKline, MedImmune, Inc. (Gaithersburg, Maryland), Novartis, Regeneron Pharmaceuticals, Inc. (Tarrytown, New York), Sanofi S.A. (Paris, France), and Teva Pharmaceuticals USA (North Wales, Pennsylvania). Spirometers used in Severe Asthma Research Program III were provided by nSpire Health, Inc. (Longmont, Colorado). 

SPIROMICS

The Subpopulations and Intermediate Outcome Measures in COPD Study (SPIROMICS) has been funded by contracts with the NIH/NHLBI (contracts HHSN268200900013C, HHSN268200900014C, HHSN268200900015C, HHSN268200900016C, HHSN268200900017C, HHSN268200900018C, HHSN268200900019C, and HHSN268200900020C) and grants from the NIH/NHLBI (grants U01 HL137880 and U24 HL141762) and supplemented through contributions made to the Foundation for the NIH and the COPD Foundation by AstraZeneca, MedImmune, Bayer Corporation (Whippany, New Jersey), Bellerophon Therapeutics (Warren, New Jersey), Boehringer-Ingelheim, Chiesi Farmaceutici S.p.A. (Parma, Italia), the Forest Research Institute, Inc. (Jersey City, New Jersey), GlaxoSmithKline, Grifols Therapeutics, Inc. (Research Triangle Park, North Carolina), Ikaria, Inc. (Hampton, New Jersey), Novartis, Nycomed Pharma GmbH (Zurich, Switzerland), ProterixBio, Inc. (Billerica, Massachusetts), Regeneron, Sanofi, Sunovion, Takeda Pharmaceutical Company (Tokyo, Japan), Theravance Biopharma, Inc. (South San Francisco, California), and Mylan N.V. (White Sulphur Springs, West Virginia). 

SHS

The Strong Heart Study has been funded in whole or in part with federal funds from the National Heart, Lung, and Blood Institute, National Institute of Health, Department of Health and Human Services, under contract numbers 75N92019D00027, 75N92019D00028, 75N92019D00029, & 75N92019D00030. The study was previously supported by research grants: R01HL109315, R01HL109301, R01HL109284, R01HL109282, and R01HL109319 and by cooperative agreements: U01HL41642, U01HL41652, U01HL41654, U01HL65520, and U01HL65521. The content is solely the responsibility of the authors and does not necessarily represent the official views of the National Institutes of Health or the Indian Health Service (IHS).

3. Regarding the Role of the Funder: 

Response: The funders had no role in study design, data collection and analysis, decision to publish, or preparation of the manuscript.

4. Regarding previously reported Competing Interests:

Response: This does not alter our adherence to PLOS ONE policies on sharing data and materials.

Response: we would like to provide the following data availability statement: 

The Researching COVID to Enhance Recovery (RECOVER) Initiative makes C4R data available to the scientific community and the public at large, allowing those interested to view the data and researchers to analyze it, learn from it, and incorporate it into future studies. BioData Catalyst (BDC) is a cloud-based research ecosystem that permits access to and a platform for the analysis of scientific data supported by NHLBI. BDC currently hosts C4R datasets from 13 of the 14 C4R cohorts; data from one cohort is not publicly available on BDC due to consent limitations. Cohort data from many of the C4R parent studies is also accessible through a dbGaP data access request. BDC provides instructions for finding and viewing C4R and cohort data as well as the steps to request authorization to use de-identified individual participant data for scientific analysis within BDC. Of note, to analyze de-identified individual participant data, researchers must receive authorization for access. This authorization is required to maintain the integrity of the data and protect participant privacy. Those wishing to access C4R data will need to submit a Data Access Request (DAR) through the database of Phenotypes and Genotypes (dbGaP). Authorized researchers can then access C4R data from BioData Catalyst® (BDC).

Response: We have added the original ethics statement to the Methods section of the manuscript and we have included a table of supervising IRBs as S2 Table. 

7. Please include a copy of Table 4 which you refer to in your text on page 27.

Response: We have relabeled Table 5 as Table 4; the original table numbering was erroneous. 

8. We note you have included a table to which you do not refer in the text of your manuscript. Please ensure that you refer to Table 5 in your text; if accepted, production will need this reference to link the reader to the Table.

Response: There are only 4 main tables. We have corrected the numbering. 

Response: we have done so. 

Response: no changes.

Additional Editor Comments:

Hello,

Thanks for submitting the manuscript to this journal. It is a well-written draft worthy of further evaluation.

Please address the comments raised by our reviewer for further evaluation

Thanks

Reviewers' comments:

Reviewer's Responses to Questions

Comments to the Author

1. Is the manuscript technically sound, and do the data support the conclusions?

Reviewer #1: Yes

2. Has the statistical analysis been performed appropriately and rigorously? 

Reviewer #1: Yes

3. Have the authors made all data underlying the findings in their manuscript fully available?

Reviewer #1: No

4. Is the manuscript presented in an intelligible fashion and written in standard English?

Reviewer #1: Yes

5. Review Comments to the Author

Reviewer #1: This manuscript by Dr Oelsner et al, “Classifying COVID-19 Hospitalizations in Epidemiology Cohort Studies: The C4R Study,” reports findings a study which used a protocolized adjudication process to co

---

## [Editor Report · Decision Letter 1]

9 Dec 2024

Classifying COVID-19 Hospitalizations in Epidemiology Cohort Studies: The C4R Study

PONE-D-24-32613R1

Dear Dr. Oelsner,

We’re pleased to inform you that your manuscript has been judged scientifically suitable for publication and will be formally accepted for publication once it meets all outstanding technical requirements.

Kind regards,

Kamal Sharma

Academic Editor

PLOS ONE

Additional Editor Comments (optional):

Hello,

The revised version of the manuscript and the reply to the reviewer in the revised version of manuscript titled "Classifying COVID-19 Hospitalizations in Epidemiology Cohort Studies: The C4R Study" is satisfactory and the revised version can now be evaluated for acceptance at the editorial desk. Thanks
---

## [Editor Report · Acceptance letter]

17 Jan 2025

PONE-D-24-32613R1 

PLOS ONE

Dear Dr. Oelsner, 

I'm pleased to inform you that your manuscript has been deemed suitable for publication in PLOS ONE. Congratulations! Your manuscript is now being handed over to our production team.

Kind regards, 

on behalf of

Dr. Kamal Sharma 

Academic Editor

PLOS ONE